# A Discussion on Different Approaches for Prescribing Physical Interventions – Four Roads Lead to Rome, but Which One Should We Choose?

**DOI:** 10.3390/jpm10030055

**Published:** 2020-06-27

**Authors:** Fabian Herold, Alexander Törpel, Dennis Hamacher, Henning Budde, Thomas Gronwald

**Affiliations:** 1Department of Neurology, Medical Faculty, Otto von Guericke University, Leipziger Str. 44, 39120 Magdeburg, Germany; 2Research Group Neuroprotection, German Center for Neurodegenerative Diseases (DZNE), Leipziger Str. 44, 39120 Magdeburg, Germany; 3German Swimming Federation, Korbacher Straße 93, 34132 Kassel, Germany; toerpel@dsv.de; 4German University for Health and Sports, (DHGS), Vulkanstraße 1, 10367 Berlin, Germany; dennis.hamacher@dhgs-hochschule.de; 5Faculty of Human Sciences, MSH Medical School Hamburg, University of Applied Sciences and Medical University, Am Kaiserkai 1, 20457 Hamburg, Germany; henning.budde@medicalschool-hamburg.de; 6Faculty of Health Sciences, Department of Performance, Neuroscience, Therapy and Health, MSH Medical School Hamburg, University of Applied Sciences and Medical University, Am Kaiserkai 1, 20457 Hamburg, Germany; thomas.gronwald@medicalschool-hamburg.de

**Keywords:** interindividual heterogeneity, dose, physical activity, physical exercise, physical training, exercise prescription

## Abstract

It is well recognized that regular physical exercise has positive effects on physical and mental health. To use the beneficial health effects of physical exercise, there are worldwide movements encouraging health care providers to include physical exercise in their care and treatments strategies. However, a crucial point in administering the “exercise polypill” is the dosing and, in turn, the prescription of the physical intervention (PI). In this perspective article, we discuss the advantages and disadvantages of different approaches to prescribe PI. In this context, we also highlight outstanding questions and potential areas of opportunity for further investigations.

## 1. Introduction

There is solid and growing evidence demonstrating that regular physical exercise exerts a substantial and positive effect on physical and mental health in all age groups because it ensures the good functioning of various organic systems, such as the metabolic system, the cardiovascular system, the musculoskeletal system, and the central nervous system [1,2,3,4,5,6,7,8,9,10,11,12,13]. Consequently, physical interventions (PI) are considered an important strategy in public health care (e.g., disease prevention and medical care). To promote this position, the “Exercise is Medicine” (EIM) initiative was established in 2007 by the American Medical Association and the American College of Sports Medicine (ACSM). This initiative aims (i) to consolidate the evidence of the positive effects of physical activity, physical exercise, and physical training on health, and (ii) to transfer this knowledge to the general public [3,14,15,16]. In reports, physical exercise is often used synonymously with physical activity, which can be misleading [17,18]. It can be said that physical exercise is always a kind of physical activity, but physical activity is not necessarily physical exercise [19]. In particular, the term *physical activity* covers all muscle-induced bodily movements leading to an increase in energy expenditure above ∼1.0/1.5 MET (metabolic equivalent of task; 1 MET = 1 kcal (4184 kJ) • kg^−1^ • h^−1^) [17,18,20,21,22,23,24]. The term *physical exercise* covers specific, planned, and structured forms of physical activity [17,18,20,21,22,23,24]. Furthermore, there is a distinction between *acute physical exercise* (i.e., a single bout) and *chronic physical exercise* (i.e., repeated bouts of acute exercises), where the latter could also be denoted as *physical training* when physical exercises (i) are conducted regularly in a planned, structured, and purposive manner and (ii) with the objective of increasing or, at least, maintaining the individual fitness and health status in one or multiple dimensions [17,18,21,22,23,24,25,26]. From this perspective, we use *physical intervention* (PI) as an umbrella term encompassing both physical exercise and physical training [22]. However, in order to use the great potential of physical exercise/physical training as a “polypill” [27,28,29,30], an appropriate prescription of the PI is crucial in order to administer an effective dose or dosage (regularly provided dose over a specific period of time) that exerts an optimal stimulus to the organism [21,31,32]. In this regard, it is assumed that an adequate exercise and/or training prescription is important for tailoring PI. We further expect that an appropriately tailored PI is likely to maximize the benefits for the single individual [21,22,31,32,33]. Indeed, there is evidence which suggests that adjusting the prescription of PI can positively influence the responsiveness of the individual person [21,33,34,35,36]. Exercise and/or training prescriptions can be used to adjust the dose of a PI. However, the terminus dose is variously defined within the literature [37].

In the recent literature, it is emphasized that external load, influencing factors, and internal load are important factors with respect to the dose [21,22]. External load is defined as the work that an individual performs regardless of internal characteristics [21,22,38,39,40,41,42,43,44,45]. Internal load is defined as individual and acute physiological, psychological, motor, and biomechanical responses to the external load and influencing factors (all environmental factors (e.g., climate, equipment) and lifestyle factors (e.g., nutrition, sleep) that can amplify or diminish the physical exercise stimuli) during physical exercise [21,22,38,39,40,41,42,43,44,45]. Both, parameters of external load (e.g., running at a speed of 12 km/h) and/or internal load (e.g., running at 74% of maximum heart rate; running at a specific rating of perceived exertion) can be used to prescribe and monitor PI. Recent approaches recommend prescribing, operationalizing, and monitoring the dose of PI by using one or multiple indicator(s) of internal load as a proxy [21,22]. However, there is currently no clear consensus on which indicator(s) of internal load is/are the most appropriate one(s) for doing so [21,46,47,48,49,50,51,52,53,54,55,56,57,58,59]. Given that an adequate exercise and/or training prescription is pivotal to administering an effective dose [21,22], this ongoing debate might produce major sources of difficulty in implementing physical exercise as medicine. Having said this, such a debate also entails great potential to serve as a promising starting point for fostering progress in both research and practical settings. To contribute to this debate, we will explain different approaches to prescribing and tailoring PI from this perspective by highlighting their potential impact on interindividual heterogeneity regarding health-related outcomes. Furthermore, we will provide recommendations for practical applications and outline potential opportunities for further investigations.

## 2. Different Approaches to Prescribing Physical Interventions

In this brief article, we will give an overview of approaches that can be used to prescribe PI. Furthermore, we will show how different approaches to exercise and/or training prescription can potentially influence the dose and outcome. In this regard, we postulate that there are four approaches to prescribe PI: approach (i) is based on the parameter(s) of external load to ensure a comparable external load across different individuals; approach (ii) is based on indicator(s) of internal load to ensure a comparable internal load across different individuals; approach (iii) aims to achieve a comparable dose across different individuals; and approach (iv) aims to achieve a comparable outcome (or a comparable change in an outcome) across different individuals (see Figure 1 and Table 1). In this regard, we define outcome (dependent variable) as a specific parameter of interest that is a function of the dose (independent variable). It is, however, worth noting that a clear assignment of a distinct variable, either to our concept of dose or to our concept of outcome, is highly context-specific. The application of those two prospective approaches (especially approach (iii), based on a comparable dose) could be promising for gaining further insights into the effects of exercise and/or training prescription on biological processes and specific outcomes. Below, however, we want to discuss all four approaches to exercise and/or training prescription in more detail.

(i) *Comparable External Load*

A prescription of PI using a comparable external load ensures that every subject exercises with the same parameters of external load (e.g., exercise prescription with each individual cycle at 60% of their peak power output for 20 min, for example, as previously determined by a performance test). This approach to exercise and/or training prescription is relatively easy to apply, but, on the other hand, it can cause, as shown in Table 1, considerable interindividual heterogeneity with respect to the internal load, the dose, and the outcome due to interindividual differences in performance capacities in various biological subsystems [44,60]. However, the exercise and/or training prescription based on a comparable external load can be favorable when the other approaches are hardly practicable and/or are not sufficiently reliable (e.g., in high-intensity interval training with relative short intervals, sprint interval training, or repeated sprint training).

(ii) *Comparable Internal Load*

By using an exercise and/or training prescription relying on a comparable internal load, it is ensured that an indicator of internal load is comparable across different individuals (e.g., an exercise prescription ensuring that each individual runs within 60% of maximum heart rate for 20 min, as previously determined by a performance test).

This approach is more favorable than approach (i), as markers of internal load take individual characteristics and environmental factors into account and reflect acute biological response(s), which are the crucial impetus for acute and chronic biological changes [21,44,57,60,61]. However, approach (ii) has also limitations: (a) we often do not know which indicator or set of indicators of internal load is/are the crucial impetus for triggering biological processes leading to specific and intended changes [21], and (b) a prescription of PI based on a specific indicator of internal load (e.g., maximum oxygen uptake or heart rate) induces a considerable heterogeneity in other indicators of internal load (e.g., level of peripheral blood lactate concentration) [50,51,52,53,62]. Therefore, this approach is likely to cause a relatively large interindividual heterogeneity with respect to the dose and the outcome (see Table 1).

(iii) *Comparable Dose*

A prescription of PI that aims to achieve a comparable dose attempts to refine an approach based on a comparable internal load (see previous paragraph (ii) comparable internal load). In this regard, one or multiple specific indicators of internal load serve as proxy of the dose and are comparable across different individuals. Here, it is a necessary to emphasize (a) that these specific indicator(s) of internal load serving as a proxy of the dose are assumed to be or are known to be (causally) involved in biological processes leading to the change in the intended outcome [21,22] and (b) that a specific internal load will be achieved by a careful and individual adjustment of external load (while taking influencing factors into account) [21]. Based on this approach, it has been hypothesized that, for instance, the peripheral blood lactate concentration could be a promising proxy for prescribing PI in exercise-cognition science, since there seems to be a neurophysiological relationship between the amount of the peripheral blood lactate concentration and neurocognitive changes [21].

The strong rationale for using specific biological responses in exercise and/or training prescription to achieve specific, but comparable stimuli across different individuals is a clear advantage and supports the use of this approach [21,46,63]. In some settings (e.g., endurance training), similar approaches are already used (e.g., approaches of threshold-based training or polarized training) [64,65,66,67], but in other settings (e.g., neuroscience), their practical implications should be more frequently considered (e.g., influence of a comparable dose on the interindividual heterogeneity of a specific neurocognitive outcome [21]). The application of a dose-related approach will lead to new insights explaining the relationships between exercise and/or training prescription, biological processes, and outcomes [21]. Such new insights could, at least in part, pave the way for an approach to prescribing PI that leads to a relatively comparable outcome across different individuals (see Table 1 and approach (iv)) and might also contribute to a refinement of the current concept of dose of PI.

(iv) *Comparable Outcome*

A prescription of PI aiming to achieve a comparable outcome (or a comparable change in an outcome) across different individuals would be the most desirable approach to exercise and/or training prescription, as it maximizes the benefit for the single individual. This approach is based on a highly tailored exercise and/or training prescription that requires a flexible adjustment of the exercise and training variables to achieve the intended outcome. Consequently, this approach leads to a relatively large interindividual heterogeneity in parameters of external load and indicators of internal load. In this case, the impact on the dose is currently relatively unclear (see Table 1). However, it should be considered that a comparable outcome (or a comparable change in an outcome) cannot be achieved in every circumstance, as individual capacities of biological subsystems (e.g., musculature) are limited (i.e., not every individual will be able to run 100 m in less than 10 s). Although it might be difficult to achieve a comparable outcome (or a comparable change in an outcome) across different individuals under some circumstances, this does not challenge the basic idea of this approach (iv) which states that a rigorous individualization of the exercise and/or training prescription (e.g., due to a flexible and individual adjustment of the external and internal load) is necessary to obtain a specific and intended outcome. With respect to health-related outcomes, an interindividual comparable outcome (or a comparable change in an outcome) is, in general, possible to obtain, as an individual optimum in a specific outcome is situated within a “normal” physiological state that can be aimed for (e.g., resting blood pressure around 120/80 mmHg [68]). In this regard, it is worth noting that approaches (i), (ii), and (iii) are outcome-oriented as well. In particular, the established approaches (i) and (ii) are based on the prescription of specific exercise and training variables of which we know (e.g., based on findings from a meta-analysis) that they lead, in general, to a certain change in an outcome parameter. However, under most circumstances, when applying approaches (i), (ii), and (iii), one would not need to consider interindividual differences in the capacities of all biological subsystems in the exercise and/or training prescription. As a result of this, these approaches do not necessarily induce comparable outcomes across different individuals, although further research is necessary to support this assumption empirically with respect to approach (iii). To come closer to a highly tailored exercise and/or training prescription as intended in approach (iv), two strategies for research in this field can be applied: (a) investigating how a comparable dose across different individuals would influence a specific outcome (see approach (iii) and Table 1) and (b) investigating how achieving a comparable outcome across different individuals would influence dose (see Table 1).

To realize strategy (b), N-of-1 studies could be a promising option. An N-of-1 study is a randomized controlled trial that is conducted in a single individual and is performed with a within-subject design with multiple crossovers [69,70,71,72,73,74,75]. In such an N-of-1 trial, the exercise and/or training prescription for the single individual needs to be systematically adjusted to change the dose and, in turn, to elucidate its effect on a specific outcome (e.g., this approach is often used in high-performance sports to determine an optimally tailored exercise and/or training prescription for an individual athlete). To draw conclusions regarding comparability and interindividual heterogeneity, such N-of-1 studies, of course, need to be conducted with several independent participants. Probably such a systematic variation of the exercise and/or training prescription in N-of-1 studies help us to better understand the underlying biological processes that lead to a specific outcome and to identify the most appropriate prescription of PI (e.g., best indicator(s) of internal load) to achieve the optimal dose and to maximize, in turn, the benefits for the individual. These theoretical assumptions, however, need to be proven empirically by performing well-designed N-of-1 studies.

In addition, the following important points are applicable to all of the above-mentioned approaches to exercise and/or training prescription and should be considered when designing and analyzing PI.

Based on their temporal characteristics, outcome should, additionally, be classified into (i) acute change in an outcome in response to a single and acute bout of physical exercise, and (ii) chronic change in an outcome in response to several exercise sessions over a distinct time period. In this context, the stimulus of acute physical exercise triggers an acute and transient physiological, psychological, motor, and biomechanical response during (e.g., transient increase of heart rate) and/or after the cessation of the single bout of physical exercise (e.g., transient increase of endocrine hormones), which leads to an acute and transient change in an outcome (e.g., transient change in blood pressure such as post-exercise hypotension). Regularly performed physical exercise over a distinct time period causes chronic response(s)/organismic adaptation(s) (e.g., basal change in endocrine hormone level or change in arterial stiffness level) which, in turn, contribute substantially to a permanent change in a distinct outcome parameter (e.g., decrease in resting systolic/diastolic blood pressure).

Furthermore, it is important to mention that wearable devices and smartphone applications offer great potential to tailor, prescribe, and monitor PI [76,77] although a stricter scientific evaluation is often needed with regard to commercially available devices/applications [78,79,80]. Notably, it has been shown that smartphone apps can positively influence exercise adherence (e.g., by providing feedback and motivation) [81,82,83], which is an important factor determining the effectiveness of a PI. However, to fully harness the great potential of wearable devices and smartphone applications in exercise and/or training prescription, further research in this direction is urgently needed [39,84].

## 3. Practical Applications

Taken together, the approach used for exercise and/or training prescription has a considerable impact on comparability across different individuals and the interindividual heterogeneity of parameter(s) of external load, indicator(s) of internal load, dose, and outcome (for an overview see Table 1). These fundamental differences between approaches to exercise and/or training prescription should be considered when designing PI or when analyzing the effects of PI (e.g., in the context of “Exercise is Medicine”). However, the most suitable and effective parameters for prescribing PI are, so far, relatively unknown [21,46,47,48,49,50,51,52,53,54,55,56,57,58,59], which, on the one hand, limits the conclusions that can be drawn with certainty, but, on the other hand, also highlights the need for further research in this direction.

From a practical and theoretical point of view, it can be unreservedly recommended (i) that, in general, specific markers of internal load should be used to prescribe and monitor PI [21,22,59], (ii) that the adaptation of internal load requires the careful and individual adjustment of external load (while taking influencing factors into account) [21], (iii) that both parameter(s) of external load and indicator(s) of internal load should be reported in detail to enhance comparability and reproducibility of PI [85], and (iv) that the most suitable exercise and/or training prescription is probably achieved by considering empirical evidence and performing interdisciplinary teamwork (e.g., integrating the perspectives of patients and the expertise of medical experts, sport scientists, and practitioners). However, the continuous improvement of the prescription of PI warrants further, arguably more critical investigation. In this regard, it is necessary to conduct further well-designed cross-sectional studies (e.g., to explore possible relationships between indicator(s) of internal load reflecting biological processes and specific outcome(s)) and longitudinal studies (e.g., to prove causality and effectiveness of such an approach) to broaden our understanding of the relationships between the exercise and/or training prescription, the underlying biological processes, and the outcome. Accordingly, we strongly advocate investigating whether PI with a comparable dose between different individuals influences the interindividual heterogeneity in distinct health-related outcomes [21,59]. Therefore, it is especially recommended to “blind” the participants by using a sham condition (e.g., active intervention with inadequate loading to induce considerable effects [86]), which would help to control for the influence of confounders (e.g., Hawthorne effect or placebo effects) [19,86,87].

## Figures and Tables

**Figure 1 jpm-10-00055-f001:**
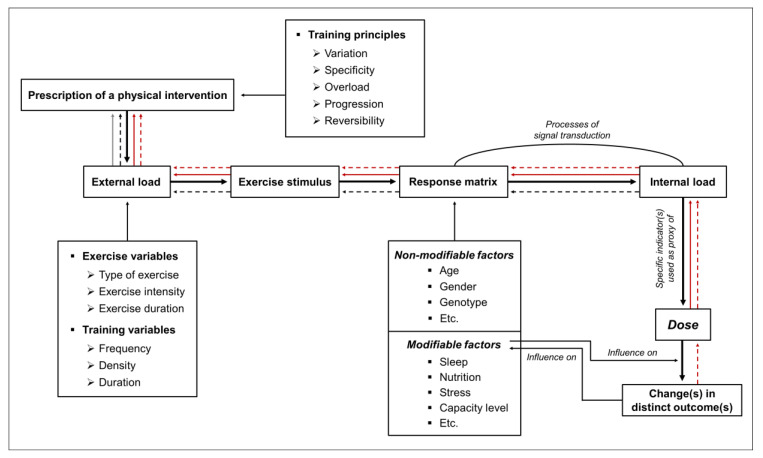
Schematic illustration of four approaches to prescribe physical interventions (PI). Approaches are based on (i) comparable external load (solid grey line), (ii) comparable internal load (dashed black line), (iii) comparable dose (solid red line), or (iv) comparable outcome (dashed red line). The prescription of PI is based on different exercise variables, training variables, and training principles inducing a specific exercise stimulus. In the so-called response matrix, the exercise stimulus interacts with the unique characteristics of the individual which leads to a measurable psychophysiological response (i.e., internal load). In short-term, a specific internal load can cause acute changes in distinct outcome parameters whereas the repetitive and long-term occurrence of an internal load can drive chronic changes in distinct outcome parameters (e.g., adaptations). Such chronic changes influence modifiable individual factors (i.e., capacity level) which, in turn, alter the biological processes in the response matrix. A detailed definition of exercise variables, training variables, and training principles, can be found elsewhere [21].

**Table 1 jpm-10-00055-t001:** Overview of approaches to prescribing physical interventions (PI) and their assumed influence on comparability across different individuals (CAI) and interindividual heterogeneity (IH) with respect to external load, internal load, dose, and outcome. ↓ indicates a relatively low comparability across different individuals or a relative low interindividual heterogeneity, whereas ↑ represents a relatively high comparability across different individuals or a relatively high interindividual heterogeneity; ? shows that we do not currently have enough knowledge to make valid assumptions. Green-shaded field: indicates a relative high compatibility across different individuals and a low interindividual heterogeneity with respect to the specific category; yellow-shaded field: illustrates a relatively low comparability across individuals and a relative high level of interindividual heterogeneity concerning the specific category; grey-shaded field: shows that it is currently not possible to make assumptions about the level of comparability across individuals and the level of interindividual heterogeneity.

	External Load		Internal Load		Dose		Outcome	
	CAI	IH	CAI	IH	CAI	IH	CAI	IH
(i) Comparable external load	↑	↓	↓	↑	↓	↑	↓	↑
(ii) Comparable internal load	↓	↑	↑	↓	↓	↑	↓	↑
(iii) Comparable dose	↓	↑	↓	↑	↑	↓	?	?
(iv) Comparable outcome	↓	↑	↓	↑	?	?	↑	↓

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
