# Peer review of "A Discussion on Different Approaches for Prescribing Physical Interventions – Four Roads Lead to Rome, but Which One Should We Choose?"

_jpm, 2020, doi:10.3390/jpm10030055_

Round 1

Reviewer 1 Report

The manuscript provides a really good approach and analysis of the considerations to take in to account to prescribe physical intervention. We know the positive effect of regular physical activity on physical and mental health, but sometimes the physical interventions and the acute and chronic load don't be controlled, so we don't knows what's really happen about the dose and respond. For this reason is so relevant a professional in sport science to control this kind of variables.   By other hand, the manuscript shows an interesting description about the actual situation, and because of this, the current study is focus on a topic of relevance and general interest to the readers of the journal. I found the paper to be overall well written and much of it to be well described. I felt confident that the authors performed a good literature review. On the other hand, I found some of the description of the paper to be well detailed, but the authors should be check the page 2, line 41 “can be misleading 1”, I think the number is a mistake, and in the summary section, the authors should be consider adding other poin about the relevance of the expert in sport cience to better desing training or exercise prescriptions. For that reason, I recommend that a minor revision is warranted.

Author Response

The manuscript provides a really good approach and analysis of the considerations to take in to account to prescribe physical intervention. We know the positive effect of regular physical activity on physical and mental health, but sometimes the physical interventions and the acute and chronic load don't be controlled, so we don't knows what's really happen about the dose and respond. For this reason is so relevant a professional in sport science to control this kind of variables.   By other hand, the manuscript shows an interesting description about the actual situation, and because of this, the current study is focus on a topic of relevance and general interest to the readers of the journal. I found the paper to be overall well written and much of it to be well described. I felt confident that the authors performed a good literature review. On the other hand, I found some of the description of the paper to be well detailed, but the authors should be check the page 2, line 41 “can be misleading 1”, I think the number is a mistake, and in the summary section, the authors should be consider adding other point about the relevance of the expert in sport science to better designing training or exercise prescriptions. For that reason, I recommend that a minor revision is warranted.

  • Thank you very much for your expert feedback. We have checked the mentioned line on page 2 and corrected this mistake. Furthermore, we have added the following sentence to incorporate the reviewers thought regarding the relevance of an expert in sport science to design an appropriate intervention: “From a practical and theoretical point of view, it can be recommended unreservedly (i) that specific markers of internal load should be used to prescribe and monitor physical interventions [20,21,56], (ii) that the adaptation of internal load requires the careful and individual adjustment of external load (while taking influencing factors into account) [20], (iii) that both, parameter(s) of external load and indicator(s) of internal load should be reported in detail to enhance comparability and reproducibility of physical interventions [82], and (iv) that probably the most suitable exercise/ training prescription is achieved by considering empirical evidence and performing interdisciplinary teamwork (e.g., integrating the perspectives of the patients, the medical experts, and the sport scientists and practitioners).”

Reviewer 2 Report

General comment

Thank you for the opportunity to review this manuscript for the Journal of Personalized Medicine. The aim of this perspective is to discuss on different approaches to prescribe physical interventions. Moreover, wants to give evidence based guidelines in order to approach correctly to exercise prescription and their assumed influence on comparability across different individuals. I really appreciate the effort of the authors working in this field, because actually exercise prescription is a trend topic in order to promote a healthy lifestyle in the general population. However, I cannot recommend this article to be accepted in the current form. In fact, several issues have to be fixed prior publication. Please see my specific comments here below.

Abstract

Line 20 and 21. Please rewrite this sentence as “It is well recognized that regular exercise has positive effects on physical and mental health”.

Introduction

Line 36 and line 41. Please check that the references in the text are in accordance with the Journal Guidelines.

Lines from 40 to 44. Please discuss here better the difference between “Physical Activity” and “Physical Exercise”. Please be consistent to use these two term appropriately in the whole manuscript.

Line 47. I suggest using “tailor physical interventions” instead of “individualize physical interventions”. Please be consistent of using it in the whole manuscript.

Lines from 53 to 61. I really appreciate that you differ from “external” and “internal” load, because these are two different parameters of training monitoring. However, I suggest adding also here the perceived load. In fact, in sport science the quantification and monitoring of training load has been the topic of many scientific works in the last fifteen years. Training load monitoring helps coaches and practitioners to individually prescribe, follow-up, analyse, adjust, and programme training sessions. To this regards the objective training parameters of external and internal training load, should be also considered with a subjective marker of training load, such as the Rating of Perceived Exertion to give a more precise quantification of training load and its perception. This is really useful for monitoring the exercise intervention in patients treated with β-blockers.

Line 71. Your introduction need to be better hypothesis driven to allow the reader to see the basis of your perspective. It also need to be clear what the practical question you are trying to address. . The key issue here is to make sure you set-up the experimental approach of your study.

Different approaches to prescribe physical interventions

Lines from 182 to 192. I suggest facing up better the concept of “Exercise Adherence”. In fact, adequate adherence to exercise is the key for the effectiveness of any training intervention. In addition, I suggest also considering the use of electronic devices to prescribe and monitor physical exercise training intervention. Actually, a number of eases of use smartphone apps that promote health behavior change are now available: most focus on health and fitness (11), and the most frequently applied behavior change techniques are goal setting, self-monitoring, and feedback on performance. To this regard, I suggest to consider here a recent paper of Bonato et al. (2020) [Bonato M, Turrini F, DE Zan V, et al. A Mobile Application for Exercise Intervention in People Living with HIV. Med Sci Sports Exerc. 2020;52(2):425‐433. doi:10.1249/MSS.0000000000002125]. In these article, the authors showed that the use of a smartphone application was functional to achieve improvements of cardiorespiratory fitness, body composition, lipid profiles and psychological outcomes. These benefits were likely associated with specific app features, i.e., providing motivation and feedback to participants and remote control for adherence to the training program prescription.

Figure 1. I suggest adding near the training principles, the parameters of volume, intensity and frequency of training. These are fundamental for programming a tailored exercise intervention both in athletes and in patients.

Line 199. I suggest changing “summary” with “Practical applications”.

Lines from 208 to 225. I suggest avoiding the use of the bulleted list. Moreover, I would suggest also inserting here the possible practical applications. What should now physicians, trainers and practitioners now have to do after reading your paper? Does it affect practice is the key factor for this section. Obviously limit speculation or qualify them, and make sure that your statement are referenced.

Author Response

General comment

Thank you for the opportunity to review this manuscript for the Journal of Personalized Medicine. The aim of this perspective is to discuss on different approaches to prescribe physical interventions. Moreover, wants to give evidence based guidelines in order to approach correctly to exercise prescription and their assumed influence on comparability across different individuals. I really appreciate the effort of the authors working in this field, because actually exercise prescription is a trend topic in order to promote a healthy lifestyle in the general population. However, I cannot recommend this article to be accepted in the current form. In fact, several issues have to be fixed prior publication. Please see my specific comments here below.

Abstract

Line 20 and 21. Please rewrite this sentence as “It is well recognized that regular exercise has positive effects on physical and mental health”.

  • We are thankful for your valuable feedback and have edited the sentence as suggested.

Introduction

Line 36 and line 41. Please check that the references in the text are in accordance with the Journal Guidelines.

  • Thank you for your careful proofreading. We have checked the mentioned text sections and corrected this mistake.

Lines from 40 to 44. Please discuss here better the difference between “Physical Activity” and “Physical Exercise”. Please be consistent to use these two term appropriately in the whole manuscript.

  • We are thankful for pointing out this shortcoming and have added the following sentences to address the reviewer’s concern.
  • “It can be said that physical exercise is always a kind of physical activity, but physical activity is not necessarily physical exercise [16]. In particular, the term physical activity covers all muscle-induced bodily movements leading to an increases in energy expenditure above ∼0/1.5 MET (metabolic equivalent of task; 1 MET = 1 kcal (4,184 kJ) × kg−1 × h−1) [14,15,17–21]. The term physical exercise covers specific, planned, and structured forms of physical activity [14,15,17–21]. Furthermore, it should be distinguished between acute physical exercise (i.e., a single bout) and chronic physical exercise (i.e., repeated bouts of acute exercises) whereas the latter could be denoted as physical training when physical exercises i.) are conducted regularly in a planned, structured, and purposive manner and ii.) with the objective to increase or, at least, maintain the individual fitness and health-status in one or multiple dimensions [14,15,17,18,20–23]. Please note that we use in this perspective physical intervention as an umbrella term encompassing both physical exercise and physical training [21].”

Line 47. I suggest using “tailor physical interventions” instead of “individualize physical interventions”. Please be consistent of using it in the whole manuscript.

  • Thank you again for your valuable feedback. We have incorporated this language advice in our manuscript.

Lines from 53 to 61. I really appreciate that you differ from “external” and “internal” load, because these are two different parameters of training monitoring. However, I suggest adding also here the perceived load. In fact, in sport science the quantification and monitoring of training load has been the topic of many scientific works in the last fifteen years. Training load monitoring helps coaches and practitioners to individually prescribe, follow-up, analyse, adjust, and programme training sessions. To this regards the objective training parameters of external and internal training load, should be also considered with a subjective marker of training load, such as the Rating of Perceived Exertion to give a more precise quantification of training load and its perception. This is really useful for monitoring the exercise intervention in patients treated with β-blockers.

  • Thank you very much for bringing this point to our attention. We completely agree with the reviewer in that point that subjectively perceived exertion (RPE) is an important parameter to quantify training load more precisely. As RPE could be seen as an indicator of internal load, we have added the following in the revised version of the manuscript to addressed this shortcoming: “Both, parameters of external load (e.g., running at a speed of 12km/h) and/or internal load (e.g., running at 74% of maximum heart rate; running at a specific rating of perceived exertion) can be used to prescribe and monitor physical interventions.”

Line 71. Your introduction need to be better hypothesis driven to allow the reader to see the basis of your perspective. It also need to be clear what the practical question you are trying to address. . The key issue here is to make sure you set-up the experimental approach of your study.

  • We are thankful for your constructive feedback. In order to address this objection, we have added the following sentences in the revised version of the manuscript: “Recent approaches recommend to prescribe, operationalize, and monitor the dose of physical interventions by using one or multiple indicators of internal load as a proxy [20,21]. However, there is currently no clear consensus on which indicator(s) of internal load is/are be the most appropriate one(s) to do so [20,43–56]. Given that an adequate exercise and/or training prescription is pivotal to administer an effective dose [20,21], this ongoing debate might evoke major sources of difficulty to implement physical exercise as medicine. Having said this, such a debate also entails great potentials to serve as a promising starting point fostering progress in both research and practical settings. To contribute to this debate, we will explain in this perspective article the different approaches to prescribe and tailor physical interventions by highlighting their potential influence on interindividual heterogeneity regarding health-related outcomes. Furthermore, we provide recommendations for practical applications and outline potential opportunities for further investigations.”

Different approaches to prescribe physical interventions

Lines from 182 to 192. I suggest facing up better the concept of “Exercise Adherence”. In fact, adequate adherence to exercise is the key for the effectiveness of any training intervention. In addition, I suggest also considering the use of electronic devices to prescribe and monitor physical exercise training intervention. Actually, a number of eases of use smartphone apps that promote health behavior change are now available: most focus on health and fitness (11), and the most frequently applied behavior change techniques are goal setting, self-monitoring, and feedback on performance. To this regard, I suggest to consider here a recent paper of Bonato et al. (2020) [Bonato M, Turrini F, DE Zan V, et al. A Mobile Application for Exercise Intervention in People Living with HIV. Med Sci Sports Exerc. 2020;52(2):425‐433. doi:10.1249/MSS.0000000000002125]. In these article, the authors showed that the use of a smartphone application was functional to achieve improvements of cardiorespiratory fitness, body composition, lipid profiles and psychological outcomes. These benefits were likely associated with specific app features, i.e., providing motivation and feedback to participants and remote control for adherence to the training program prescription.

  • We are thankful that the reviewer has pointed out this shortcoming. To address her/his concerns we have added the following in the revised version of that paragraph: ”Furthermore, it is important to mention that wearable devices and smartphone applications offer a great potential to tailor, prescribe and monitor physical interventions [73,74] although a stricter scientific evaluation is often needed with regard to commercially available devices/applications [75–77]. Notably, it has been shown that smartphone apps can positively influence exercise adherence (e.g., by providing feedback and motivation) [78–80] which is an important factor determining the effectiveness of a physical intervention. However, to fully harness the great potential of wearable devices and smartphone applications in exercise and/ or training prescription, further research in this direction is urgently needed [36,81].”

Figure 1. I suggest adding near the training principles, the parameters of volume, intensity and frequency of training. These are fundamental for programming a tailored exercise intervention both in athletes and in patients.

  • Thank you very much for this thoughtful suggestion. We have revised Figure 1 and the training variables has now been added in more detail.

Line 199. I suggest changing “summary” with “Practical applications”.

  • Thank you again for your expert feedback. We have edited the heading as suggested.

Lines from 208 to 225. I suggest avoiding the use of the bulleted list. Moreover, I would suggest also inserting here the possible practical applications. What should now physicians, trainers and practitioners now have to do after reading your paper? Does it affect practice is the key factor for this section. Obviously limit speculation or qualify them, and make sure that your statement are referenced.

  • We are thankful for the reviewer’s thoughtful comments and have edited the mentioned text section as follows: “Taken together, the used approach of exercise and/or training prescription has a considerable impact on the comparability across different individuals and interindividual heterogeneity of parameter(s) of external load, indicator(s) of internal load, the dose, and the outcome (for overview see Table 1). These fundamental differences between the approaches of exercise and/or training prescription should be considered when designing physical interventions or when analyzing the effects of physical interventions (e.g., in the context of “Exercise is Medicine”). However, the most suitable and effective parameters to prescribe physical interventions are, so far, relatively unknown [20,43–56] which, on the one hand, limits the conclusion which can be drawn with certainty, but, on the other hand, also highlights the need for further research in this direction. From a practical and theoretical point of view, it can be recommended unreservedly (i) that specific markers of internal load should be used to prescribe and monitor physical interventions [20,21,56], (ii) that the adaptation of internal load requires the careful and individual adjustment of external load (while taking influencing factors into account) [20], (iii) that both, parameter(s) of external load and indicator(s) of internal load should be reported in detail to enhance comparability and reproducibility of physical interventions [82], and (iv) that probably the most suitable exercise and/or training prescription is achieved by considering empirical evidence and performing interdisciplinary teamwork (e.g., integrating the perspective of medical experts, sport scientists, and patients). However, the continuous improvement of the prescription of physical interventions warrants further, arguably more critical investigations. In this regard, it is necessary to conduct further well-designed cross-sectional studies (e.g., to explore possible relationships between indicator[s] of internal load reflecting biological processes and specific outcome[s]) and longitudinal studies (e.g., to prove causality and effectiveness of such an approach) to broaden our understanding of the relationships between the exercise and/or training prescription, the underlying biological processes, and the outcome. Accordingly, we strongly advocate to investigate whether physical interventions with a comparable dose between different individuals influence the interindividual heterogeneity in distinct health-related outcomes [18,56]. Thereto, it is especially recommended to “blind” the participants by using a sham condition (e.g., active intervention with inadequate loading to induce considerable effects [83]) which helps to control for the influence of confounders (e.g., Hawthorne effect or placebo effects) [16,83,84].

Reviewer 3 Report

This article is written well, so I recommend to publish this after revision.

1) Figure 1 will need to revise more clearly, because it is hard to see and understand for readers.

2) Table 1 and explanations are also not clearly, so I recommend to revise these too.

Author Response

This article is written well, so I recommend to publish this after revision.

1) Figure 1 will need to revise more clearly, because it is hard to see and understand for readers.

  • We thank the reviewer for her/his feedback. We have simplified Figure 1 in the revised version of the manuscript.

2) Table 1 and explanations are also not clearly, so I recommend to revise these too.

  • Thank your again for constructive criticisms. We have added a more detailed explanation of the Table 1 in the table heading.

Round 2

Reviewer 2 Report

Thank you very much for addressing my previous comments. I think the paper is much improved, and you should be commended for your thorough approach to the editing process.

Author Response

We want to take this opportunity to thank the reviewer for her/his constructive feedback throughout the review process as her/his feedback helped us a lot to improve the overall quality of our manuscript.